# Ti(C, N) as Barrier Coatings

**Katarzyna Banaszek [1] and Leszek Klimek [2,]*** 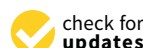

[1] Department of General Dentistry, Medical University of Lodz, Pomorska Str. 251, 92-213 Lodz, Poland
[2] Institute of Materials Science and Engineering, Lodz University of Technology, Stefanowskiego Str. 1/15, 90-924 Lodz, Poland
* Correspondence: leszek.klimek@p.lodz.pl

**Abstract:** Metals and their alloys are materials that have long been used in stomatological prosthetics and orthodontics. The side effects of their application include reactions of the body such as allergies. Their source can be corrosion products as well as metal ions released in the corrosion process, which penetrate the surrounding tissue. In order to prevent the harming effect of metal alloys, intensive research has been performed to purify metal prosthetic restorations by way of modifying their surface. The study presents the investigation results of Ti(C, N)-type coatings applied to alloy Ni–Cr by means of the magnetronic method. Five coatings differing in the nitrogen and carbon content were investigated. The studies included the determination of the coatings' chemical composition, construction, as well as the amount of ions released into the environment: distilled water, 0.9% NaCl and artificial saliva. The performed investigations showed that, in reference to an alloy without a coating, each coating constitutes a barrier reducing the amount of ions transferred into the examined solutions. So, Ti(C, N)-type coatings can be considered for biomedical applications as protective coatings of non-precious metal alloys.

**Keywords:** Ni–Cr alloy; Ti(C, N) coatings; ion release

## 1. Introduction

Metals and their alloys are materials that have been used in stomatological prosthetics and orthodontics for a long time now. The side effects of their use can be some reactions of the body, e.g., allergies. Their source can be the corrosion products as well as the metal ions released in the corrosion process, which penetrate the surrounding tissue. Metals and basic metal alloys under the conditions of the oral cavity do not exhibit a hazardous effect as long as they are corrosion resistant. However, the exceptionally strong corrosive properties of the biological environment make all the basic metal alloys unable to resist corrosion [1,2]. Often, prosthetic restorations require the use of several types of metal alloys, which additionally contribute to the creation of electrogalvanic elements and intensifies the corrosion speed [3]. The process of corrosive destruction is accompanied by the penetration of hazardous alloys, corrosion products, and/or metal ions into the environment [4]. The metal ions, released during the corrosion process from the metal restoration constructions, are transferred into the digestive system and accumulate in the stomach, liver, spleen, bones, and mucous membrane. The metal ions released from metallic implants accumulate in the surrounding tissues, often reaching very high concentrations. After exceeding the critical concentration, toxic as well as allergic reactions take place [5]. The degree of the harmful effect of prosthetic metal constructions depends mainly on the amount of released corrosion products and/or metal ions (being in direct relation with the restorations' proneness to corrosion, as well as their size and the conditions in which they exist, e.g., the presence of another metal in the oral cavity, the pH, etc.) and also on the degree of their harmful (toxic) effect [6]. One should remember, however, that the hazardous effect of prosthetic

restorations is also a result of the cooperation of many other factors, such as the galvanic currents or mechanical poling.

The experimental studies demonstrated a hazardous effect of the basic metal alloys [7–10]. The results of the clinical tests proved also a harmful effect of prosthetic constructions made of the basic metal alloys [11]. In stomatology, usually, we observe local toxicity of metals and their alloys. It is thought that, among many metal alloys used in clinical applications, NiCr and CoCr alloys characterize in the highest degree of hazardous effect [4,6,12–14]. This is also confirmed by clinical observations of gums in contact with crowns made of NiCr alloys, compared to precious metal alloys [15].

More and more often, one can observe allergic reactions to metal alloys used in stomatological prosthetics and orthodontics. Hypersensitivity to metals causes inflammatory reactions [16], in the case of patients both with skeletal prostheses and those made of CoCr alloys, as well as crowns made of CrNi steels, NiCr alloys, and alloys based on palladium and orthodontic devices [11,17]. So, the clinical application of metal alloys in prosthetics poses the risk of the occurrence of an allergic reaction.

Despite the corrosion and the harmful effect of metal alloys, they continue to be applied for long-term construction elements of prosthetic restorations [18], being the most frequently used despite the fact that new "metal-free" technologies are being introduced [19,20], together with full-ceramic restoration materials [21] and constructions based on zirconium oxide [22,23]. In order to prevent the hazardous effect of metal alloys, intensive research has been performed aiming at the development of fully biocompatible materials, i.e., such that undergo both a mechanical and functional integration with the tissue. These studies are concentrated on:

- The development of metal alloys with a higher corrosion resistance;
- The purification of metal prosthetic restorations through their surface modification.

In recent years, more and more often, layers applied by various methods, e.g., chemical vapor deposition (CVD), Physical vapor deposition (PVD), and sol-gel, have been used for that purpose. Among the many coatings obtained by these methods, the most frequently applied ones include metal carbides, oxides and nitrides [24–29]. A special attention should be paid to titanium carbides and nitrides. This mainly results from their high strength and corrosion resistance. Research works have been performed aiming at modifying the technology of obtaining nitride layers in order to improve their properties, which depend, among other things, on the $TiN/Ti_2N$ ratio in the layer [30]. Another research direction is obtaining layers made of titanium carbonitride Ti(C, N). As it was shown by the preliminary investigations [31–34], Ti(C, N) layers demonstrate a better corrosion resistance and wear resistance, and they significantly reduce the amount of the released metal ions and so, they can potentially be used as coatings for prosthetic and orthodontic metal restorations.

The tests performed on Ti(C, N) coatings applied on Ni–Cr alloys showed that they exhibit the proper mechanical and physicochemical properties, and also they are much less toxic than Ni–Cr alloys [34–37]. The toxicity reduction is probably connected with the barrier effect of the coating, consisting in a reduction of the amount of substrate ions being transferred into the environment. So, the aim of the study became a comparison of the amount of ions released into the solutions in samples with Ti(C, N) coatings with different carbon and nitrogen contents with samples without a coating.

## 2. Materials and Methods

The test material was constituted by samples made of NiCr Heraenium NA by Heraeus Kulzer (Hanau, Germany), alloy in the form of cylinders, 8 mm in diameter and 10 mm high, for the chemical composition and microscopic tests (Figure 1a), as well as bars cast from the same alloy, 2 mm in diameter and 45 mm long – for the ion release studies (Figure 1b). The initial composition of the alloy determined by the X-ray fluorescent analysis method with the use of an SRS300 spectrometer by SIEMENS (Munich, Germany) has been given in Table 1.

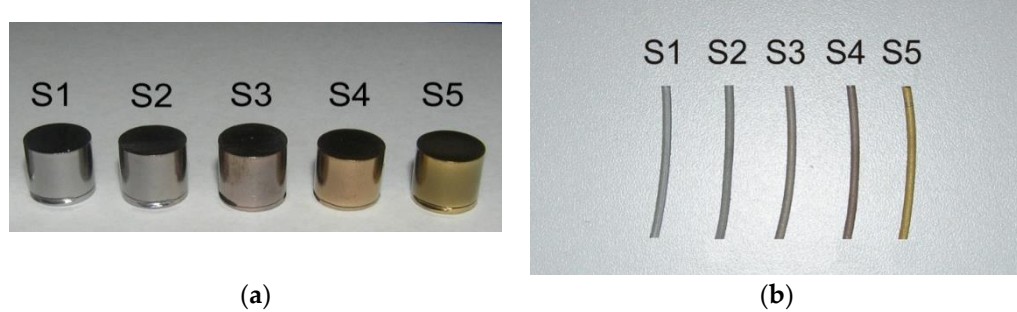

**Figure 1.** Test samples: (**a**) microscopic and chemical composition tests; (**b**) ion release tests.

**Table 1.** Chemical composition of tested alloy.

| Element Percentage wt.% | | | | | | |
|---|---|---|---|---|---|---|
| **Cr** | **Mo** | **Si** | **Fe** | **Co** | **Mn** | **Ni** |
| 24.79 | 8.89 | 1.57 | 1.33 | 0.17 | 0.12 | residue |

The disks were divided into five groups, according to the content of carbon and nitrogen. The coatings were deposited by the method of magnetron sputtering. The samples were cleaned in a detergent, i.e., acetone, and next, after they were placed in the chamber, they underwent ionic cleaning. The ionic cleaning was carried out with the use of an ionic gun applying argon as the working gas. The pressure in the chamber was 0.0021 Pa. The samples prepared in this way were coated with layers by means of the magnetron sputtering method. In order to improve the adhesion of the Ti(C, N) layers, first, an adhesive sublayer made of pure titanium was applied for the time of 120 s, with the argon pressure of 0.24 Pa and the following parameters of the magnetron's work: 3 kW/around, 4.5 A. After 2 min, a reactive gas was slowly released: nitrogen, acetylene or a mixture of the two. The deposition time of the main coating was the same for all the processes and equalled 7200 s. The constant voltage polarization during the deposition was −100 V. During the deposition, the samples were moved above the target's surface in a swinging motion, in order to homogenize the thickness. In each case, the pressure of the main process equalled 0.27 Pa. The type of reactive gases and their flows have been given in Table 2. These were the only varying parameters of the processes.

**Table 2.** Reactive gas flow.

| Gas | Flow Unit | Samples | | | | |
|---|---|---|---|---|---|---|
| | | **S1** | **S2** | **S3** | **S4** | **S5** |
| $N_2$ | [sccm] | 0 | 4 | 8 | 12 | 16 |
| $C_2H_2$ | | 8 | 6 | 4 | 2 | 0 |

After the deposition, the vacuum chamber with the samples was cooled down and only then the charge was removed.

The obtained samples underwent chemical composition tests performed on the coatings as well as microscopic observations made on cross-sections. The assessment of the chemical composition of the coatings was performed by means of an optical emission spectrometer with a glow discharge spectrometer (GDS) 850 A by LECO (St. Joseph, MI, USA), and with an alternating current lamp RF (with radio frequency) with a 4 mm diameter anode. The microscopic tests on the samples' cross-sections were performed in a scanning electron microscope S-3000N (HITACHI, Tokyo, Japan). The obtained images of the coatings have been presented in Figure 2.

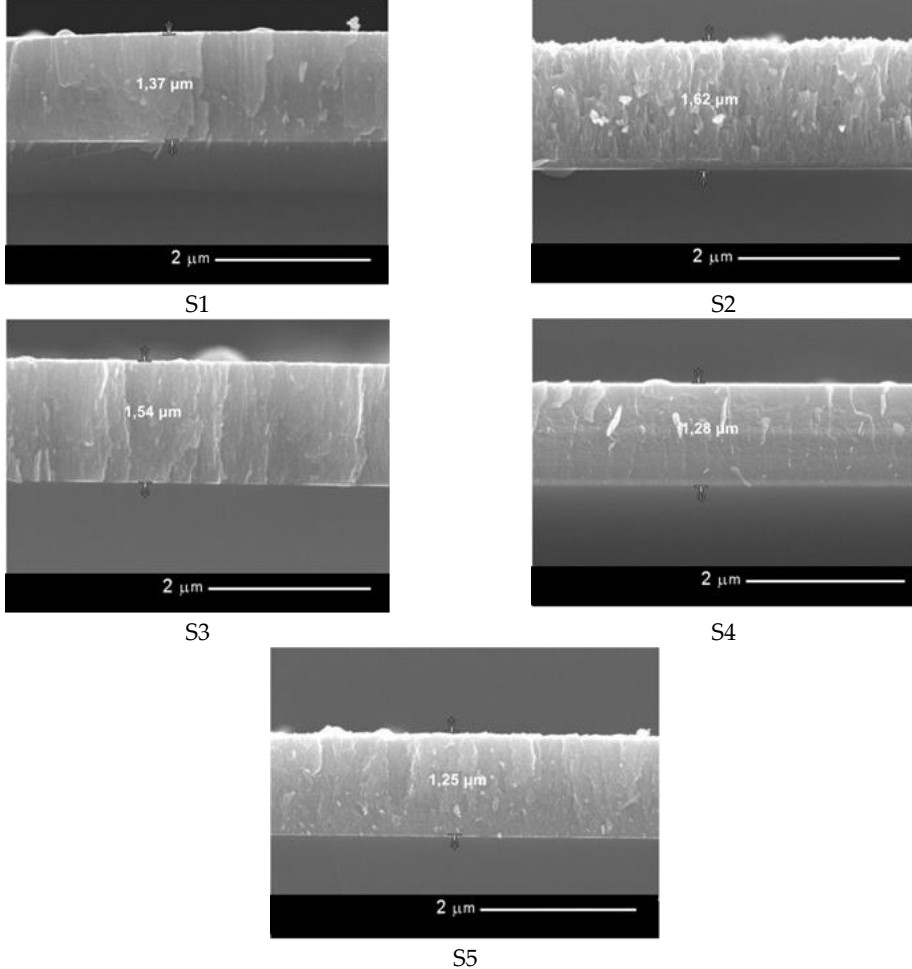

**Figure 2.** Cross-sections of particular coatings.

The ion release tests were carried out in the following environments:

- Distilled water;
- Physiological saline solution (0.9% NaCl in $H_2O$);
- Artificial saliva according to FusayamaMayer (2 $dm^3$ distilled water, 0.8 g NaCl, 0.8 g KCl, 1.59 g $CaCl_2 \cdot 2H_2O$, 1.56 g $NaH_2PO_4 \cdot 2H_2O$, 0.01 g $Na_2S \cdot 9H_2O$ and 2 g urea) [38].

The metal bars, 2.0 mm in diameter and 45 mm long, made of alloy NiCr, with Ti(C, N)-type coatings as well as without any, were placed in polystyrene vessels, volume 30 mL. In each vessel, a mixing element was placed. Each container was filled with 10.0 mL of the liquid. The analyzed systems were incubated at 36.7 ± 0.2 °C. In each environment, the tests were carried out for 10, 30 and 90 days. After the completion of the given cycle, three 3 mL samples were collected from each vessel and an analysis by means of an inductively coupled plasma optical emission spectrometer ICP OES Optima 8300 by PERKIN ELMER (Waltham, MA, USA) was performed. The content of the following ions was examined: Ni, Cr, and Mo. The spectrometer calibration (determination of the calibration curves) was performed by means of standard water solutions with the concentration of the determined elements of 0.5, 1.0, 5.0, and 10 mg/L. The measurement of the released ions was made directly on solutions without dilution or mineralization of the sample. The concentrations of the examined elements were calculated in reference to the prepared analytical curves. The element concentration in each sample was measured twice. Two analytical lines were applied for each measurement, as shown in Table 3. The application of different analytical lines was aimed at identifying the possible spectral interferences.

The presented results constitute an average of three measurements from two analytical lines. For each measurement, about 1.5 mL solution was used.

**Table 3.** Analytical lines used for the measurement.

| Element | Analytical Lines [nm] | |
|---------|-----------------------|---------|
| Ni | 232.003 | 341.476 |
| Cr | 267.716 | 357.869 |

The mass $X_j$ of the ions released since the beginning of the process was calculated from the following formula:

$$X_j = C_j \times V_c \tag{1}$$

where:

- $X_j$—mass of the ions released from the sample [mg];
- $C_j$—spectrometrically determined ion concentration [mg/dm$^3$];
- $V_c$—volume of the liquid in which the samples were submerged [dm$^3$].

The mass of the ions released from the sample per surface unit, was calculated from the following formula:

$$M_j = X_j/S_p \tag{2}$$

where:

- $M_j$—mass of the released ions per surface unit [mg/mm$^2$];
- $X_j$—mass of the ions released from the sample [mg];
- $S_p$—sample surface in contact with the liquid [mm$^2$].

The obtained results concerning the amount of the released ions were subjected to a statistical analysis by means of the non-parametrical Mann Witney U test [39]. Statistical analyses were conducted using PQStat statistical software (version 1.6.4.122).

## 3. Test Results

Chemical composition of deposited Ti(C, N) layers is shown in Table 4.

**Table 4.** Chemical composition of tested coatings.

| Coating | Element Percentage at.% | | | Element Percentage wt.% | | |
|---------|-------|-------|-------|-------|-------|-------|
| | Ti | C | N | Ti | C | N |
| S1 | 53.50 | 48.50 | 0.00 | 80.18 | 19.82 | 0.00 |
| S2 | 52.91 | 33.91 | 13.80 | 79.51 | 13.90 | 6.60 |
| S3 | 51.94 | 28.22 | 19.84 | 78.76 | 11.67 | 9.57 |
| S4 | 47.78 | 20.05 | 32.17 | 75.26 | 8.61 | 16.12 |
| S5 | 56.79 | 0.00 | 53.21 | 79.78 | 0.00 | 20.22 |

The effect of the performed process was obtaining coatings with the thickness from 1.3 to 1.6 μm and a diversified chemical composition, from pure titanium carbide with the content of about 48 at.% C, to pure titanium nitride with the content of about 53 at.% N. Additionally, three intermediate layers with the contents of about 34 at.% C and about 14 at.% N, about 28 at.% C and about 20 at.% N, about 20 at.% C and 32 at.% N were obtained. The total content of carbon and nitrogen in the coatings varied within the scope of 47% to 52%.

Figure 2 shows the microscopic images of the samples' cross-sections together with the particular coatings and their thicknesses (SE contrast).

The microscopic observations made it possible to evaluate the coatings' thickness, which equalled 1.25 to 1.62 μm. The coatings were air-tight, uniformly distributed and without defects. On the fractures of the coatings, one can see their columnar construction, where, besides some small differences in thickness, a difference in the crystallites' shape can also be observed. The coatings from groups S2, S3, S4 seem to have a more fine-grained structure.

Figures 3–8 show the amount of released nickel and chromium ions for the particular samples referred to a sample's surface unit.

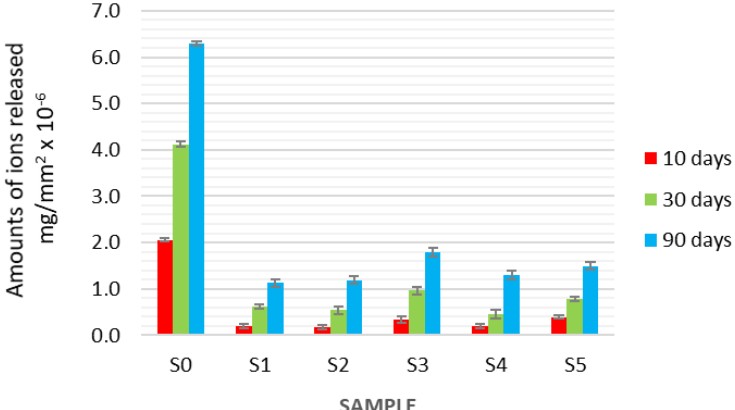

**Figure 3.** Amounts of ions Ni released into distilled water for the particular coatings after 10, 30, and 90 days.

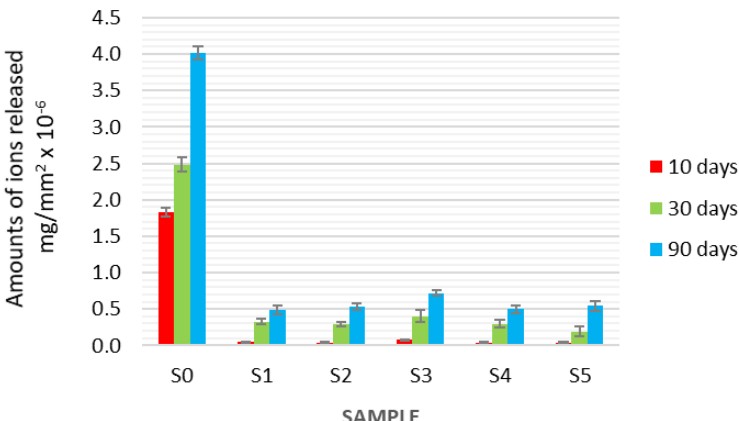

**Figure 4.** Amounts of ions Cr released into distilled water for the particular coatings after 10, 30, and 90 days.

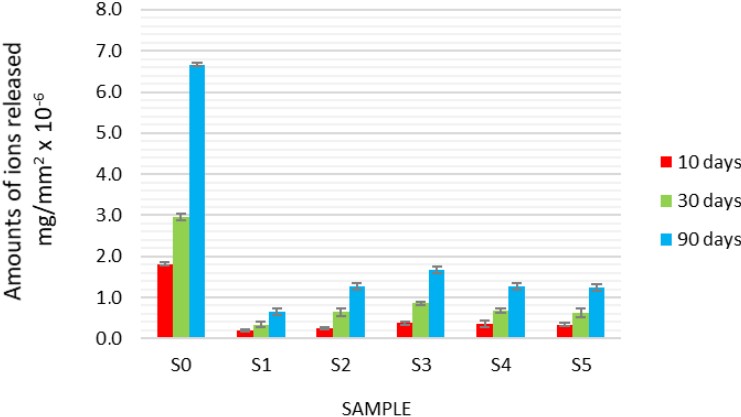

**Figure 5.** Amounts of Ni ions released into 0.9% solution NaCl for the particular coatings after 10, 30, and 90 days.

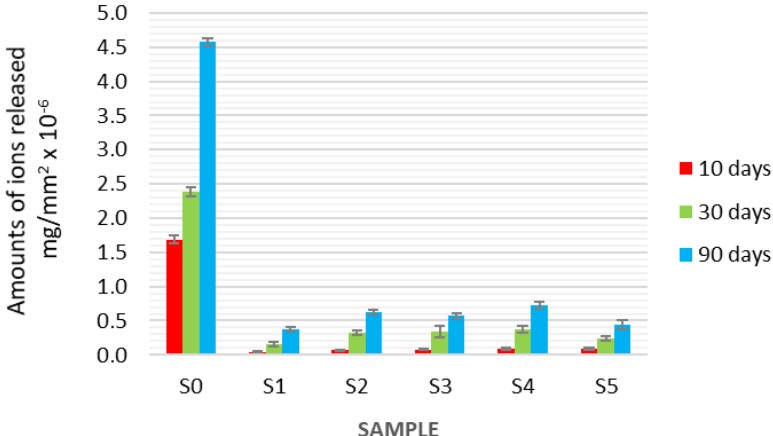

**Figure 6.** Amounts of Cr ions released into 0.9% solution NaCl for the particular coatings after 10, 30, and 90 days.

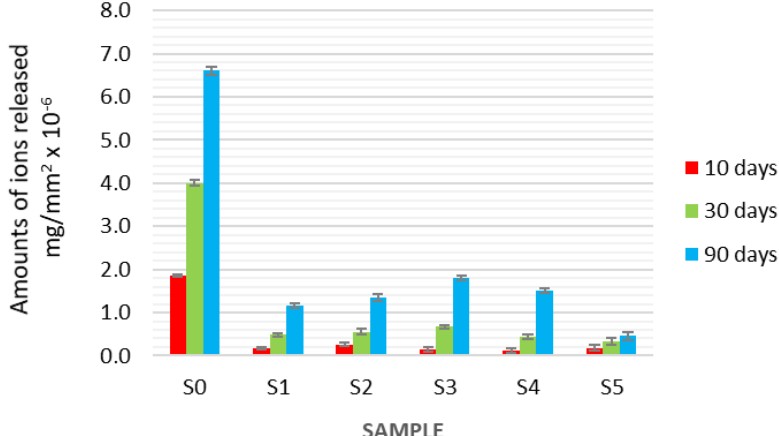

**Figure 7.** Amounts of Ni ions released into artificial saliva for the particular coatings after 10, 30, and 90 days.

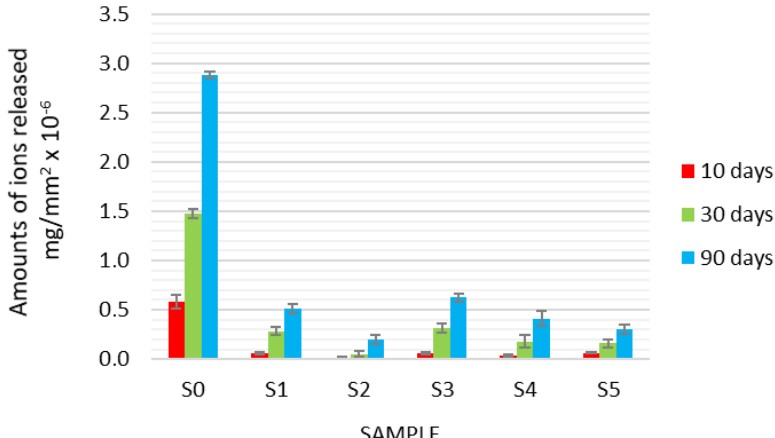

**Figure 8.** Amounts of Cr ions released into 0 artificial saliva for the particular coatings after 10, 30, and 90 days.

*3.1. Examinations of the Amount of Released Ions*

In the case of the control group S0, for each examined environment, an increase of the amount of released ions in respect of the results obtained after 10 days was observed—after 30 days about a

double increase, after 90 days about a triple increase. In the case of the coated samples, the dynamics of the ion release was as follows.

### 3.2. Distilled Water Environment

In the environment of distilled water, after 10 days, in the samples covered with the studied coatings, a decrease of the released ions was recorded in reference to the uncoated alloy Ni–Cr (group S0)—from 5 to 12 times lower for the nickel ions and about a dozen times lower for the chromium ions. The lowest amount of ions was released from sample S4 (Ni ions—about 10× less, Cr—about 60× less). In turn, in reference to sample S0, the largest amount of ions was released from the sample with coating S3 (Ni ions—about 6× less, Cr—about 26× less).

After 30 days, in distilled water, about 6 to 10 times less nickel ions and about 3 to 8 times less chromium ions were recorded in reference to the non-coated alloy Ni–Cr (group S0). The lowest amount of ions was released from the samples in group S4, both in the case of nickel and chromium. The largest amount was recorded for group S3 and S5 (TiN), i.e., about 4.5 and about 6 times less nickel ions, and 6 and 3 times less chromium ions, respectively, in comparison with group S0.

In the environment of distilled water, after 90 days, in reference to the non-coated alloy Ni–Cr (group S0), the lowest amount of ions was released from the samples in group S1 and S4 (about 6 and about 5 times less for the Ni ions, respectively, and about 8 times less chromium ions, in comparison to the samples of pure alloy Ni–Cr (S0)). In the case of the remaining coatings, the decrease was about 3–4 times for the nickel ions and about 6–8 times for the chromium ions. The largest amount of nickel ions was released from samples S2 and S3, i.e., about 3 and 3.5 times less, whereas for chromium—6 and 8 times less, in respect of the non-coated alloy in group S0.

### 3.3. 0.9% NaCl Environment

In the 0.9% NaCl environment, after 10 days, in the samples covered with the tested coatings, a significant decrease of the released ion amount in reference to the non-coated alloy Ni–Cr (group S0) was recorded—from 4 to 10 times lower for the nickel ions and over a dozen times lower for the chromium ions. The lowest amount of ions was released from samples S1 (Ni ions—about 10× less, Cr—about 40× less). In turn, in reference to sample S0, the largest amount of ions was released from sample S3 (Ni ions—about 5× less, Cr—about 20× less). The amounts of released Cr ions in samples S2, S4 and S5 were at a similar level to those in the case of S3.

After 30 days, in the 0.9% NaCl environment, about 4 to about 9 times less nickel ions and about 6 to about 15 times less chromium ions was recorded in reference to the non-coated alloy Ni–Cr (group S0). The lowest amount of nickel and chromium ions was released from S1 (nearly 9 times and 15 times less, respectively). The largest amount of nickel ions was released from groups S3, S2, S4 (about 4 times less than in the non-coated group). With regard to the chromium ions, the largest amount was released in group S4—about 4 times less compared to group S0. In group S5, the amount of the released nickel ions was slightly more advantageous than in groups S3, S4 and S2, while in the case of the released chromium ions in group S5, the result was much better than in groups S2, S3, and S4, i.e., 10 times less compared to group S0.

In the 0.9% NaCl environment, after 90 days, in reference to the non-coated alloy Ni–Cr (group S0), the lowest amount of ions was released from the samples in group S1 (about 10 times less for the Ni ions and about 12 times less for the chromium ions in reference to the samples of pure alloy Ni–Cr). In the case of the remaining coatings, the decrease equalled about 4 times for the nickel ions and about 6–10 times for the chromium ions. The amount of the released nickel ions in groups S2, S4 and S5 was at a similar level, i.e., about 5 times less, and in S3—about 4 times less. Chromium was released in the largest amounts from sample S4, i.e., 6× less in reference to the non-coated alloy in group S0. In the remaining groups, i.e., S2, S3, and S5, the results were similar and the amount of the released chromium ions decreased about 7, 8 and 10 times, respectively, in reference to the alloy without a coating.

*3.4. Artificial Saliva Environment*

In the environment of artificial saliva, after 10 days, in the samples covered with the examined coatings, a significant decrease of the released ions was recorded in reference to the non-coated alloy Ni–Cr (group S0), which was from 5 to 11 times lower for the nickel ions and from 9 to 45 times lower for the chromium alloys. In reference to group S0, the Ni ion release was as follows: 11 times in group S1, 7.5 times in group S2 and about 5 times in groups S3, S4 and S5. The lowest amount of chromium, in reference to the samples without coatings, was released in group S2, i.e., 45 times less, whereas in group S4—over 14 times less. In groups S3 and S5, about 10 times less and in S5—9 times less Cr ions were recorded.

After 30 days, in the artificial saliva environment, from about 6 to about 12 times less nickel ions and from 5 to about 30 times less chromium ions were recorded in reference to the non-coated alloy Ni–Cr (group S0). The lowest amount of nickel ions was released from the samples in group S5 (12 times less compared to the non-coated alloy), whereas the lowest amount of chromium was released from sample S2 (about 30 times less in reference to the non-coated alloy). The highest amount of Ni and Cr ions was released from the samples in group S3, where 6 and 4.5 times less ions was released compared to the non-coated alloy. In groups S2, S1 and S4, 7×–9× less nickel ions were released than in the group without coating. With regard to the chromium ions in groups S4 and S5, 8–9 times less released ions were recorded, while in group S1—about 5 times less (close to that in group S3) compared to group S0.

In the artificial saliva environment, after 90 days, the results for the released Ni and Cr ions in the examined groups were diversified. The amounts of released Ni ions were from 3.5 to 14.5 times lower in respect of the non-coated alloy Ni–Cr, whereas in the case of the chromium ions—from 4.5 to 14 times lower. The least nickel ions was released from the samples in group S5 (about 14.5 times less), and the least chromium ions in group S2—14 times less. The highest amount of released ions was observed in group S3—3 times less nickel ions and 4 times less chromium ions in respect of the group without coatings.

## 4. Discussion

The amounts of the released nickel and chromium ions are different, which, among other reasons, results from the content of these elements in the alloy (about twice as much nickel as chromium). However, the ratio of the released nickel ions to the chromium ions is about 3 to 4 times higher. So, one should presume that the examined coatings constitute a better barrier for chromium than for the nickel. This can result from the difference of the size of the nickel $Ni^{2+}$ and chromium $Cr^{3+}$ ions, which equals about 20%.

The performed tests showed that, in the case of the Ti(C, N)-type coating, the amounts of Ni and Cr ions released into each of the tested environments was much lower than in the case of the alloy without a coating. In the case of each environment, the statistical analysis of the obtained results demonstrated a statistically significant decrease in the amount of released ions in the coated samples in respect of the samples without a coating. No statistically significant differences in the amount of released ions were observed between the samples covered with the particular coatings. The lack of significant differences in the penetration of the ions should not be surprising; the coatings have a similar construction and corrosion resistance. However, a more thorough analysis of the results makes it possible to notice a certain tendency showing that, in most cases, the amount of ions penetrating the coatings from groups S2, S3, and S4 is a little higher than that of the ions penetrating the coatings from groups S1 and S5, and this difference results from their thickness, as coatings S2 and S3 are the thickest and so, they should constitute the best barrier. This can be explained in two ways. First, these coatings are more fine-grained. Their fine-grained structure can be demonstrated by the results of diffraction tests included in the study [31] While the results presented there refer to the phase identification of coatings, as well as their texture and stresses, in the analysis of the width of the reflexes from phases Ti(C, N) (samples S2, S3, S4), we observe their slight widening in respect of phases TiC (S1) and TiN



(S5), which can prove their more fine-grained structure. A fine-grained structure significantly increases the amount of grain boundaries, and these, as we know, are the areas of facilitated diffusion. So, nickel and chromium from the alloy can penetrate the coating more easily. Also, one should take into account the fact that these processes take place at ambient temperatures, where other diffusion mechanisms are hindered. Another reason for such a behaviour of the coatings can be their crystallographic structure. Carbide TiC has a similar structure of the crystal lattice as that of nitride TiN, and they differ only in the parameters of the lattice. In the carbonitrides being formed, the carbon atoms are successively replaced by the nitrogen atoms (the latter are placed in the position of the former, without a change in the crystal lattice). As carbon atoms are bigger than nitrogen atoms, this change causes a certain relaxation of the lattice, which can create a situation of facilitated diffusion.

In the analysis of the amount of ions released into the particular examined solutions, no statistically significant differences were established, yet certain tendencies could be observed. The highest amount of ions released into the environment of a 0.9% NaCl solution results from the aggressiveness of the chlorine ions. Chloride ions have a clear electronegative character and they favour the metals' passing from the atomic state into the ionic state. This is especially visible in the alloy without a coating. The amounts of the ions released from the pure metal are slightly higher, both in the environment of the 0.9% NaCl solution and artificial saliva, as both solutions contain chlorine ions. The slightly lower amount of ions released into distilled water can be explained by the diffusion into a non-saturated environment. The lowest amount of ions from the coated samples were released into the solution of artificial saliva, which might be explained by the fact that the latter contains phosphates and sulfides, which can form low-soluble phosphates and sulfides with the substrate metals, and they, in turn, can block the discontinuity of the structure and the porosity of the coating, thus hindering the passing of the ions.

Covering metal alloy elements with coatings can significantly reduce the cytotoxicity of these elements, thus providing the possibility of their use in contact with various tissues of the human body. The investigations performed at many scientific centres have confirmed the positive results of the use of different coatings. In a study concerning the application of diamond like carbon (DLC) coatings on metal elements, the authors demonstrated this in short-term and long-term investigations in vivo performed on rat males, breed WISTAR. The modified samples were implanted under the skin for 24 h, 1, 4 and 26 weeks. After that time, the animals were put down and the tissues surrounding the modified elements and the spleen were collected. The evaluation of two routine dyed pieces of tissue and spleen showed that: 1) the carbon coating protects the tissue surrounding the metal implants against metal ion penetration, 2) no immunological reaction of the tissue or the spleen occurs, whereas the non-coated elements cause a strong reaction, 3) the statistically important differences (pb0.05) refer to the number and type of the cell elements and the degree of reaction in respect of the modified and non-modified elements [40]. Also, studies of similar coatings were performed, where the obtained amounts of released metal ions were correlated with the estimated biological response of the cell viability test (osteoblast cell line Saos-2) and the bacterial colonization test (strain Escherichia coli DH5$\alpha$). The results showed that depositing hybrid DLC-type coatings by means of the magnetron sputtering radio frequency plasma assisted chemical vapour deposition (RF PACVD/MS) method makes it possible to obtain an air-tight coating, which prevents diffusion of harmful elements from the metallic substrate [41].

A reduced toxicity level is also exhibited by nitride and carbide coatings. Jang et al. studied biological Ti samples covered with TiN and 3-1-2,5-diphenyl tetrazolium bromide test (MTT) coatings, among other things, by examining their vitality by means of the MTT test after 8 days of being placed in the medium. The test showed a higher survivability of the cells from the group coated with TiN (by 125%) and TiAlN (by 117%) compared to the pure Ti group (100%) [42]. In the studies of Bramm et al., titanium was coated with titanium carbide TiC in the pulsed laser deposition (PLD) technology and evaluated in vitro in 3 cell lines as well as in vivo in respect of the integration with the bone, in tests performed on animals. The homogeneous TiC coatings covering titanium had a positive effect on the

bones forming the cells, both in vitro and in vivo. According to the authors, these effects result from many factors, including both the chemistry of the coating and its morphology, with the consideration of the proper roughness. The TiC coatings simultaneously increase the implant's hardness and protect titanium from the aggressive attacks of the body fluids and tissues. The authors conclude that the use of TiC coatings on titanium improves the biocompatibility; also, with the appropriate roughness, it stimulates the proliferation of osteoblasts, their adhesion and diversity, as well as improves the osteointegration of the implant [43].

The biological tests performed on Ni–Cr alloys covered with the presented coatings also showed an improvement of the biocompatibility of the coated alloy in reference to the non-coated alloy. The evaluation by means of the MTT viability test performed on human microvascular endothelial cells coming from the skin—HMEC-1—showed that, in the case of the samples incubated for 24 and 96 h, we can observe statistically significant differences (with the value $p < 0.05$) between the non-coated samples and all the other samples coated with Ti (C, N) [36]. Also, the investigations with the use of fibroblasts proved that no Ti(C, N) coating affects the activity of the oxidoreductive enzymes, ether in the case of the direct cell culture or the application of extracts of the tested materials [37]. Such a behaviour of the Ti(C, N)-type coatings has also been observed by other researchers. Serro et al. [44] compared the cytotoxicity of 48-h extracts from implants coated with carbonitride Ti(C, N) (containing 21.6% C) and nitride TiN on mouse fibroblasts. Examining the morphology and biological activity and comparing them with the cells bred in a normal medium, they did not observe a cytotoxic effect of Ti (C, N) and TiN coatings on those cells. In the studies concerning a TiN coating on a nickel alloy [45], it has been established that it does not cause a cytotoxic irritation; a significant increase of the gum fibroblast on the surfaces of nickel alloys coated with TiN has been recorded, as opposed to the surface of the alloys without a TiN coating.

It can be inferred from the performed studies that there are no significant differences in the amounts of the ions which have passed through the particular coatings. Also, their biological properties are similar [36,37]. So, it is impossible to point to the best one. During clinical proceedings, their properties should rather be considered with regard to a particular application. As it was demonstrated in earlier studies, they exhibit big differences in hardness, modulus of elasticity and shear resistance [34].

The positive effect of the coatings is caused by various factors. Undoubtedly, a decrease in the harmful ions has basic importance. Ti(C, N)-type coatings additionally prevent oxidation, which, as is suggested by Toshifumi, could have a significant effect on the surface wettability [46]. This parameter is important, e.g., during the cells' adhesion to the substrate. The wettability measurements of the presented coatings demonstrated their higher hydrophilicity than alloy Ni–Cr [34]. Also, Li confirmed that NiTi coated with TiN exhibits a much higher hydrophilicity than in the case when no coating is applied. Additionally, he pointed to a significant role of the surface roughness [47], which can be especially important during the osteointegration of the implants with the bone [48].

Correlating the results of the amounts of the ions released from alloys with biological responses, we can see that the presence of the coatings causing a reduction of the amount of the ions released into the environment clearly favours a reduction of the alloys' cytotoxicity.

## 5. Conclusions

- All the examined Ti(C, N)-type coatings significantly limit the amount of the metal ions released into the environment.
- Ti(C, N)-type coatings can be considered for biomedical applications as protective coatings for non-precious metal alloys.
- Because there are no statistically significant differences in the amounts of the ions released by the particular coatings, during the selection of the latter, their other properties should be taken into consideration.

**Author Contributions:** Conceptualization, K.B. and L.K.; Methodology, L.K.; Validation, L.K. and K.B.; Formal Analysis, L.K.; Investigation, L.K.; Resources, L.K.; Data Curation, K.B.; Writing—Original Draft Preparation, K.B.; Writing—Review and Editing, L.K.; Visualization, K.B.; Supervision, L.K.; Funding Acquisition, K.B. and L.K.

**Funding:** This research received no external funding.

**Conflicts of Interest:** The authors declare no conflict of interest. We declare that the research is disclosed all conflicts interest statement, or explicitly state that there are none.

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
