# Peer review of "Ti(C, N) as Barrier Coatings"

_coatings, doi:10.3390/coatings9070432_

Reviewer 1 Report

The manuscript is focused on the influence of Ti(C,N) coating as a protective layer against the release of Cr and Ni ions from the alloy into selected solutions.  It is shown that the layer can be considered for biomedical applications as protective coatings for non-precious metal alloys.

1. Although the manuscript contains several interesting results, their analysis is not provided.

The whole manuscript is written rather as a research report.

2. Chapter ‘Results’ contains only tables and figures without any comment.

3. The first part of the discussion is just a description of results, the second part  is only a continuations of the introduction. Any deeper analysis of results is missing.

Formal errors:

4. use decimal point instead of comma everywhere

5. use ‘chemical composition’ instead of ‘Element percentage’ in tables

6. 119 (0,9% NaCl w H 2 O)-→ (0,9% NaCl in H 2 O)

7. 144 -145 dcm3 ??

8. 154 the non-parametrical Mann Witney U test  â†’ probably provide a reference

9. 159 give the information that the image was taken in SE contrast

10. Fig. 3 – Fig. 8: remove  redundant decimal places on the scale of the y-axis

11. 165 – 171  48% at. C  â†’    about 48 at.% C  etc.

Author Response

Thank you for your observations.

1. Although the manuscript contains several interesting results, their analysis is not provided.

Reply: Corrected. The discussion was extended and the conclusion added

2. The whole manuscript is written rather as a research report.

Reply: Corrected. The discussion was extended and the conclusion added

3. Chapter ‘Results’ contains only tables and figures without any comment.

Reply: In the chapter TEST RESULTS, there are no comments, as we assumed the discussion to be included in the TEST RESULTS AND DISCUSSION section, which we did, at the same time, expanding the analysis of the results.

4. The first part of the discussion is just a description of results, the second part is only a continuations of the introduction. Any deeper analysis of results is missing.

Reply: Corrected. The discussion was extended and the conclusion added。

Formal errors:

5. Use decimal point instead of comma everywhere

Reply: Corrected

6. Use ‘chemical composition’ instead of ‘Element percentage’ in tables

Reply: With regard to the legends of the tables, The Reviewer suggests the use of "chemical composition" in the titles of the tables, while the data included in them refer to the percentage of the particular elements, and so, in our opinion, they should remain as they are.

7. 119 (0,9% NaCl w H2O)→(0.9% NaCl in H2O)

Reply: Corrected

8. 144 -145 dcm3 ??

Reply: Corrected dm3

9. 154 the non-parametrical Mann Witney U test  â†’ probably provide a reference.

Reply: Corrected

10. 159 give the information that the image was taken in SE contrast.

Reply: Corrected

11. Fig. 3 – Fig. 8: remove  redundant decimal places on the scale of the y-axis.

Reply: Corrected

12. 165 – 171  48% at. C  â†’    about 48 at.% C  etc.

Reply: Corrected

Reviewer 2 Report

1. The imformations of this study are very deneficial to the patients, when they are treated with metal prothetic restrations.

The article is almost acceptable except for minor correction.

 In Materials and Methods

2. P4 line 125 : Incubation temperature is 36,7+0.2℃, what this temperature is reflecting?

3. P4 line154 : Mann Witney U test. U should be italic.

 In Discussion

4. P8~P9 line 248

 These contents are should be in result section.

5. Why the thickness of coating is different from each other and does the thickness influence the release of metal ions ?

Author Response

Thank you for the suggestions concerning the expansion of the Reference section. We referred only to the first article. The other article concerns an entirely different group of materials and it is difficult to be referred to.

Reviewer 3 Report

1. The manuscript is well done and performed. Prosthetic materials are often the cause of allergies and in this paper it is interesting to read the results. nickel is often used in dental prostheses, therefore it would be interesting to add bibliographical references regarding the dental prosthesis. I suggest adding the following references: 

Bioactive titanium surfaces: Interactions of eukaryotic and prokaryotic cells of nano devices applied to dental practice 10.3390/biomedicines7010012

Interface between MTA and dental bonding agents: Scanning electron microscope evaluation 10.4103/jispcd.JISPCD_521_16

Author Response

1. The manuscript is well done and performed. Prosthetic materials are often the cause of allergies and in this paper it is interesting to read the results. nickel is often used in dental prostheses, therefore it would be interesting to add bibliographical references regarding the dental prosthesis. I suggest adding the following references: 

Bioactive titanium surfaces: Interactions of eukaryotic and prokaryotic cells of nano devices applied to dental practice 10.3390/biomedicines7010012

Interface between MTA and dental bonding agents: Scanning electron microscope evaluation 10.4103/jispcd.JISPCD_521_16

Thank you for your comments. According to the suggestions, we have expanded the discussion by explaining the mechanisms which may cause changes in the amounts of the released ions.

Reviewer 4 Report

Good day,

Please see my comments/suggestions that are embedded in the attached pdf file of your manuscript. 

I highly recommend some re-ordering of your manuscript, as a significant part of your discussion section is more appropriate for the introduction.  In addition, I recommend some additional discussion of your results to explain some possible mechanisms for the observed reductions in each fluid... In its present form, your discussion section appears to be a listing of the result, with very little analysis.  It does not appear that you attempted to answer the reason why these coatings performed as they did... further analyses would not only improve the discussion section, but would provide additional support to your very brief conclusions.

Author Response

1.   Thank you for your comments. According to the suggestions, we have expanded the discussion by explaining the mechanisms which may cause changes in the amounts of the released ions.

2. Were these cast as well, or commercially procured?

Alloy Heraenium NA by Heraeus Kulzer, corrected in paper.

3. Casting parameters, heat treatments, etc?  can you provide some particulars on the manufacturing process? 

Pre-made cylinders, manufacturer does not share parameters.

4. This sentence should really be moved to the next paragraph, which describes the coatings... if left here, it implies that each as-prepared cylinder/bar has a vayring amount of carbon and nitrogen in the bulk.

Coorected in paper.

5. How was the power applied: DC, pulsed DC, HiPIMS, RF?

Pulsed DC with frequency 150 kHz. This information was not given in the next, as we considered it too detailed.

6. any effects from target poisoning during the deposition? 

Parameters of the process were constantly monitored in order to prevent the poisoning.

7. how thick is your Ti adhesion layer? how far did C,N diffuse into this layer during the deposition? 

During deposition process Ti(C,N) is created on the surface right away, there is no diffusion of Ci N to Ti.

8. color images would be better.

Corrected. According to reviewer’s suggestion the images are colouful

9. typo... should read "1.62" not "1, 62"

Corrected

10. too definitive from just a single cross-section... there were no observed defects.

Typical photo of a cross-section was included in the article. Research was conducted on 5 different samples observed in numerous places of cross-sections.

11.  why?  there should be some discussion of the results rather than just a listing of the reduction of ion-release into the different fluids...

some additional information, such as surface roughness and an estimate of coating porosities would be very helpful in elucidating the reason behind the reduction in ion-release... this would lend itself to some quantitative discussion of why more Ni than Cr, other than there is more Ni than Cr in the base alloy.

Discussion of results and an explanation attempts are presented in the following part of the article. Roughness of all the samples was similar, therefore it did not influence the results. Microscopic observations did not show coating porosities.

The content of Ni in the alloy is approximately double of the content of Cr. Amount of released Ni ions is 3-4 times as big. It may result from the difference of their size.

12. This information (line 260 through 315) would be more suitable for the introduction rather than in the discussion section, as it primarily covers previous results that are relevant, but do not directly support (or explain) your results. 

It can be moved. We put it in that place, because it shows clinical effects that can be associated with releasing of ions. So I want keep it in its current form.

Round  2

Reviewer 1 Report

1. Manuscript describes a positive influence of Ti(C,N) coatings on a reduction of toxic ions release into the environment. The format of the manuscript is rather non-standard, a combination of chapter 3 and 4 into a single chapter Results and discussion is recommended.

The manuscript still contains numerous typographic errors:

2. p. 51 : [4, 6 12 – 14] → [4, 6, 12 – 14]

3. p 100 120s, → 120 s, 0.24Pa → 0.24 Pa leave space between the number and the unit also in the rest of the text.

4. p 166-170 use everywhere at. % and not % at, %  or at %!

5. p 258 about 20%. → about 20 %.

6. p288 of a 0,9% NaCl → of a 0.9 % NaCl

7. p 292 of the 0,9% NaCl → of the 0.9 % NaCl

8. p 320-321 why the text is underlined

9. p 349 properties are similar []. → give the reference

10. 380-491: Check carefully for numerous typing errors and use journals standard abbreviations including dots, use only one standard format of journals citations.

11. p { margin-bottom: 0.25cm; line-height: 120%; }

Author Response

We agree with comments of the reviewer.

1. Manuscript describes: a positive influence of Ti(C,N) coatings on a reduction of toxic ions release into the environment. The format of the manuscript is rather non-standard, a combination of chapter 3 and 4 into a single chapter Results and discussion is recommended.

In the chapter TEST RESULTS, there are no comments, as we assumed the discussion to be included in the TEST RESULTS AND DISCUSSION section, which we did, at the same time, expanding the analysis of the results. With regard to the legends of the tables, The Reviewer suggests the use of "chemical composition" in the titles of the tables, while the data included in them refer to the percentage of the particular elements, and so, in our opinion, they should remain as they are. The other remarks of the Reviewer have been considered and the text has been edited accordingly.

The manuscript still contains numerous typographic errors:

2. p. 51 : [4, 6 12 –14] → [4, 6, 12 – 14]

Corrected

3. p 100 120s, → 120 s, 0.24Pa → 0.24 Pa leave space between the number and the unit also in the rest of the text.

Corrected

4. p 166-170 use everywhere at. % and not % at, %  or at %!

Corrected

5. p 258 about 20%. →about 20 %.

Corrected

6. p288 of a 0,9% NaCl→ of a 0.9 % NaCl

Corrected

7. p 292 of the 0,9%NaCl → of the 0.9 % NaCl

Corrected

8. p 320-321 why the text is underlined

Corrected

9. p 349 properties are similar []. → give the reference

Corrected

10. 380-491: Check carefully for numerous typing errors and use journals standard abbreviations including dots, use only one standard format of journals citations.

Corrected

11. p { margin-bottom: 6.25px; line-height: 120%; }

The manuscript was corrected according to guidelines.